# Multivariable Panel Data Cluster Analysis of Meteorological Stations in Thailand for ENSO Phenomenon

Porntip Dechpichai , Nuttawadee Jinapang, Pariyakorn Yamphli, Sakulrat Polamnuay, Sittisak Injan and Usa Humphries *

Department of Mathematics, Faculty of Science, King Mongkut's University of Technology Thonburi,
126 Pracha Uthit Rd., Bang Mot, Thung Khru, Bangkok 10140, Thailand; porntip.dec@kmutt.ac.th (P.D.);
nuttawadee.jina@mail.kmutt.ac.th (N.J.); pariyakorn.tuan@mail.kmutt.ac.th (P.Y.);
sakunrat.pol@mail.kmutt.ac.th (S.P.); sittisak_injan@hotmail.com (S.I.)
* Correspondence: usa.wan@kmutt.ac.th; Tel.: +66-2470-8822

**Abstract:** The purpose of this research is to study the spatial and temporal groupings of 124 meteorological stations in Thailand under ENSO. The multivariate climate variables are rainfall, relative humidity, temperature, max temperature, min temperature, solar downwelling, and horizontal wind from the conformal cubic atmospheric model (CCAM) in years of El Niño (1987, 2004, and 2015) and La Niña (1999, 2000, and 2011). Euclidean distance timed and spaced with average linkage for clustering and silhouette width for cluster validation were employed. Five spatial clusters (SCs) and three temporal clusters (TCs) in each SC with different average precipitation were compared by El Niño and La Niña. The pattern of SCs and TCs was similar for both events except in the case when severe El Niño occurred. This method could be applied using variables forecasted in the future to be used for planning and managing crop cultivation with the climate change in each area.

**Keywords:** Euclidean distance timed and spaced; meteorological station; multivariable panel data cluster analysis

## 1. Introduction

In the past, the climate in Thailand was largely influenced by monsoon winds, such as southwest moonsoon and northeast moonsoon, resulting in Thailand having a predominantly rainy season and dry season (summer and winter) taking place at a relatively certain time. Currently, however, there has been an El Niño–La Niña phenomenon known as the ENSO phenomenon (ENSO) that affects the climate. The ENSO phenomenon is caused by variations in the Southern Hemisphere's climate system. It is a phenomenon that has a connection between ocean phenomena and ocean winds. It brings about climatic variations, causing unusually high rainfall and unusual drought [1]. There are three types of weather variability: drought, rain and cold disasters, and tropical cyclones. Thailand's proximity to the Western Pacific makes it directly affected by El Niño during 1997–1998, which resulted in drought, lower than normal rainfall, and higher than normal air temperatures across the country [2]. In 1999–2000, during the La Niña period, Thailand experienced more rainfall than usual and cold weather, breaking records in many provinces [2]. Thailand is in the humid tropics, which is suitable for agriculture. Most of its population is engaged in agriculture, so agricultural products are the main source of the country's income and, therefore, vital to its economy. The 12th Agricultural Development Plan (2017–2021) summarizes the agricultural situation in terms of climate change and seasonal variability, resulting in decreased agricultural productivity. Existing plant species are unable to adapt to changing climate conditions, especially the ongoing drought from 2012 to 2015, damaging important crops. This may be due to insufficient observation or experience by farmers to cope with unprecedented situations in time, posing a risk of loss of productivity and increased pro-

duction costs [3]. ENSO-related climate variability exerts strong influences on agricultural production in different regions, including in Thailand [4–9].

Cluster analysis, unsupervised learning, have been applied in many studies to define spatial and temporal variability from climate variables. In previous studies, only one variable, mostly focusing on rainfall in a time series format, has been used for spatial and temporal cluster [10–12]. However, there are other climate factors that affect agricultural production such as relative humidity and temperature, which statistically significantly affected sugarcane production, which was likely to decrease in the year of El Niño and to increase in the year of La Niña [13]. Although there are some studies which employed longitudinal meteorological factors such as rainfall, air temperature, humidity, pressure, wind, evaporation, etc., they firstly average data over the time into the general cross-sectional data and then the distance between samples is calculated for clustering [14]. Averaging over the time will result in a high amount of data loss because the mean shows the average change in the data, yet it does not show the distribution of the data [15–18].

It would be beneficial to study variation across different geographic scales using multivariable panel hierarchical clustering from ENSO-effected climate variables in Thailand, obtained from the conformal cubic atmospheric model (CCAM). There are seven weather variables, including rainfall, average temperature, highest temperature, lowest temperature, temperature difference from highest temperature, temperature difference from lowest temperature, relative humidity, and solar radiation according to the locations of the weather stations of the Thailand Meteorological Department. These monthly data have been characterized by a combination of panel data, cross-sectional data, and time-series data representing behavioral units and periods.

Therefore, this research will employ the distance measurement that does not need to average the data, which is Euclidean distance timed and spaced, to cluster meteorological weather stations in Thailand and discover the seasonal pattern for each cluster using climate factors associated with precipitation when ENSO phenomena occur, since changes in rainfall are important variables affecting agricultural productivity. The studied method, cluster analysis on multivariable panel data with climate change application, therefore, could be applied to the future data from weather models to group area and season. The clustering framework applied in this study is shown in Figure 1. The results could be used as a guideline to benefit the agricultural sector or the relevant agencies to prepare for the upcoming changes resulting from climate change. In addition, spatial and timely management plans can also be appropriately executed, including drought monitoring, water management of both agricultural areas, as well as crop management.

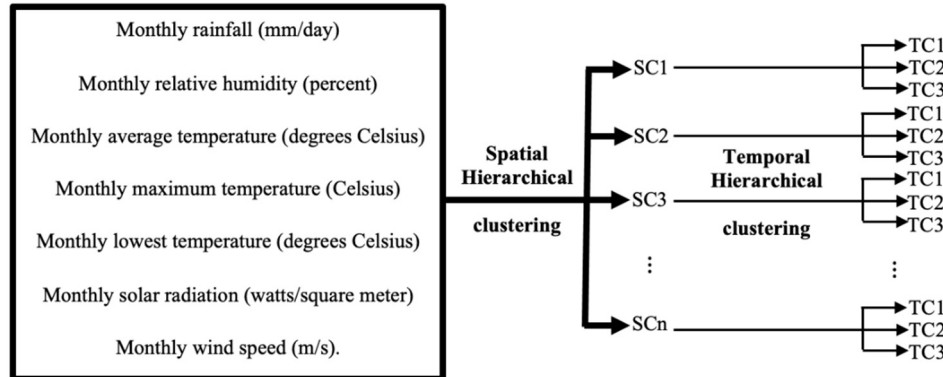

**Figure 1.** The multivariable panel data clustering framework.

## 2. Materials and Methods

### 2.1. Study Area

Thailand is located between latitudes 5°37′ N and 20°27′ N and longitudes 97°22′ E and 105°37′ E. A total of 124 stations of the Thai Meteorological Department (Figure 2) were selected for the cluster analysis.

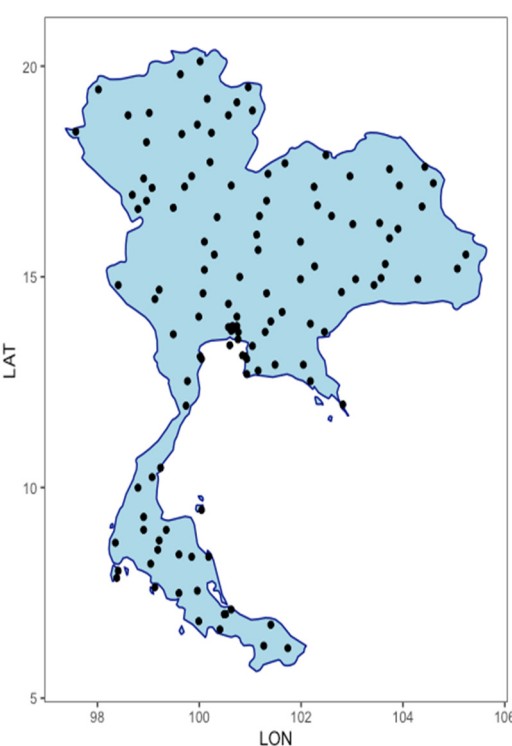

**Figure 2.** Spatial distribution of 124 meteorological stations in Thailand from the Thai Meteorological Department (TMD).

### 2.2. El Niño–Southern Oscillation (ENSO)

El Niño–southern oscillation (ENSO) is a periodic change in the oceanic atmosphere system in the tropical Pacific Ocean that affects climate around the world. It occurs every three to seven years (average five years) and typically lasts nine months to two years, associated with floods, droughts, and other global disturbances. During normal or non El Niño conditions, trade winds blow west across the Pacific Ocean. The western part of the equatorial Pacific is characterized by warm, wet, and low-pressure weather conditions due to the accumulation of moisture in the form of typhoons and thunderstorms.

During the ENSO event, there was an increase in air pressure across the Indian Ocean, Indonesia, and Australia, and a decrease in air pressure over Tahiti and the rest of the central and eastern Pacific Ocean. The trade winds in the South Pacific weaken or head east, and warm water spreads eastward from the western Pacific and Indian Ocean to the eastern Pacific. This has led to widespread droughts in the western Pacific and dry eastern Pacific rainfall. While El Niño is characterized by unusually warm ocean temperatures in the central to eastern Pacific Ocean, La Niña is characterized by unusually cold ocean temperatures in the region, but warmer waters in the western Pacific Ocean, as shown in Figure 3. However, as El Niño conditions lasted for several months, more global warming occurred in the oceans.

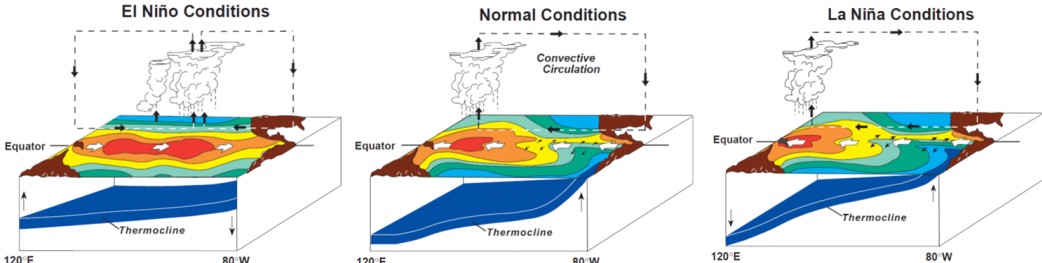

**Figure 3.** Model of surface temperature, wind, area of rising air and thermoline (blue surface) in the tropical Pacific during El Niño, Normal, and La Niña (https://reefresilience.org/th/stressors/climate-and-ocean-change/el-nino-southern-oscillation/, accessed on 24 March 2022).

In this study, the Oceanic Niño Index (ONI) from the National Oceanic and Atmospheric Administration (2020) was used to identify the El Niño–southern oscillation. The ONI is the 3-month running mean of the sea surface temperature anomaly in the Niño 3.4 region (5° N–5° S, 120°–170° W). The ONI index exceeding +0.5 °C or −0.5 °C for at least five consecutive months was considered as a full-fledged El Niño (E) or La Niña (L). According to Null report, the three latest very strong El Niño events (ONI ≥ 2 °C) in 1982, 1997, and 2015 and three latest strong La Niña events (−1.5 to −1.9 °C) in 1999, 2007, and 2011 were selected to study the climate variations [19].

### 2.3. Conformal Cubic Atmospheric Model (CCAM)

The CCAM is a dynamic global climate model developed by the Commonwealth Scientific and Industrial Research Organization (CSIRO), Division of Atmospheric Research, Australia. It is used to forecast global climate through dynamic scale reduction by generating a grid covering the region's forecast area [20]. The model has also been developed by adding physical parameterization schemes that include longwave radiation, shortwave radiation, aerosol, cumulus convection, cloud distribution, soil temperature, etc., to reduce the climate forecast error. The CCAM dataset was downscaled to 10 km grid resolution, which is sufficient for the analysis of both spatial and temporal forecasts at the regional level [21,22]. Data were changed from grid data to station format, which covers 124 meteorological measurement stations across Thailand (Figure 1).

Climate variables, focusing mainly on agricultural-related variables for cluster analysis, were used in this study. They consist of a total of 7 variables: rainfall (mm/day), relative humidity (percent), average temperature (degrees Celsius), maximum temperature (degree Celsius), minimum temperature (degrees Celsius), solar radiation (watts/square meter), and wind speed (m/s). Monthly data of those variables were collected for the years 1987, 1999, 2000, 2004, 2011, and 2015, of which the ENSO phenomenon occurred.

### 2.4. Multivariate Panel Data

Panel data is the combination of cross-sectional data and time-series data representing behavioral units over the time $(x_{ij}(t))$. Data were collected from cross-section data, which collects the value of the variables in each unit at a given point in time. Then, the data were repeatedly collected from the same unit at a subsequent time, either yearly, quarterly, monthly, weekly, daily, or hourly. If each panel unit is observed at the same time point, a data set is called balanced panel data. Consequently, if a balanced panel contains $n$ panel units and $T$ periods, the number of observations in the dataset is necessarily $N = n \times T$. However, if at least one panel unit is not observed every period, a data set is called unbalanced panel data. Therefore, the number of observations in the unbalanced panel dataset is $N < n \times T$.

Multivariate panel data has a very complex structure and cannot be represented by a simple two-dimensional table. Table 1 shows the multivariate combination of data in a two-dimensional table format, where $n$ represents the number of samples collected, $p$ represents the number of variables $(x_1, x_2, \ldots, x_p)$, $T$ represents the length of time and represents the

data value of the $i$th sample and $j$th variable at time $t$, where $i \in [1, n]; j \in [1, p]; t \in [1, T]$. Descriptive statistics, such as mean and variance of $j$th variable, is calculated as Equations (1) and (2), respectively [15–18].

$$\overline{x}_j = \frac{1}{T} \frac{1}{n} \sum_{t=1}^{T} \sum_{i=1}^{n} x_{ij}(t), \tag{1}$$

$$s_j^2 = \frac{1}{T} \sum_{t=1}^{T} \frac{1}{n-1} \sum_{i=1}^{n} \left[ x_{ij}(t) - \overline{x}_j(t) \right]^2, \tag{2}$$

**Table 1.** Multivariate panel data format for spatail cluster.

| Sample ($i$) | Time Index ($t$) | | | |
|---|---|---|---|---|
| | 1 | 2 | $\cdots$ | $T$ |
| | $x_1, x_2, \ldots, x_p$ | $x_1, x_2, \ldots, x_p$ | $\cdots$ | $x_1, x_2, \ldots, x_p$ |
| 1 | $x_{11}(1), x_{12}(1), \ldots, x_{1p}(1)$ | $x_{11}(2), x_{12}(2), \ldots, x_{1p}(2)$ | $\cdots$ | $x_{11}(T), x_{12}(T), \ldots, x_{1p}(T)$ |
| 2 | $x_{21}(1), x_{22}(1), \ldots, x_{2p}(1)$ | $x_{21}(2), x_{22}(2), \ldots, x_{2p}(2)$ | $\cdots$ | $x_{21}(T), x_{22}(T), \ldots, x_{2p}(T)$ |
| $\vdots$ | $\vdots$ | $\vdots$ | $\cdots$ | $\vdots$ |
| $n$ | $x_{n1}(1), x_{n2}(1), \ldots, x_{np}(1)$ | $x_{n1}(2), x_{n2}(2), \ldots, x_{np}(2)$ | $\cdots$ | $x_{n1}(T), x_{n2}(T), \ldots, x_{np}(T)$ |

The values for monthly climate variables were organized in two configuration matrices. Matrix $N \times p$ had monthly data ($T$) for stations ($n$) in its rows ($N = n \times T$) and the variables ($p$) in the columns. It was used to identify clusters of similar stations. Furthermore, monthly climate variables within these clusters ($N_c$) were analyzed to discover seasonality within the spatial cluster. For the second step, monthly climate variables were arranged in $T \times N_c$ rows, and the variables ($p$) were set up in columns (Table 2).

**Table 2.** Multivariate panel data format for temporal cluster.

| Month ($t$) | Station Index ($i$) | | | |
|---|---|---|---|---|
| | 1 | 2 | $\cdots$ | $N_c$ |
| | $x_1, x_2, \ldots, x_p$ | $x_1, x_2, \ldots, x_p$ | $\cdots$ | $x_1, x_2, \ldots, x_p$ |
| Jan | $x_{11}(1), x_{12}(1), \ldots, x_{1p}(1)$ | $x_{11}(2), x_{12}(2), \ldots, x_{1p}(2)$ | $\cdots$ | $x_{11}(N_c), x_{12}(N_c), \ldots, x_{1p}(N_c)$ |
| Feb | $x_{21}(1), x_{22}(1), \ldots, x_{2p}(1)$ | $x_{21}(2), x_{22}(2), \ldots, x_{2p}(2)$ | $\cdots$ | $x_{21}(N_c), x_{22}(N_c), \ldots, x_{2p}(N_c)$ |
| $\vdots$ | $\vdots$ | $\vdots$ | $\cdots$ | $\vdots$ |
| Dec | $x_{121}(1), x_{122}(1), \ldots, x_{12p}(1)$ | $x_{121}(2), x_{122}(2), \ldots, x_{12p}(2)$ | $\cdots$ | $x_{121}(N_c), x_{122}(N_c), \ldots, x_{12p}(N_c)$ |

### 2.4.1. Multivariate Cluster Analysis

Cluster analysis is an unsupervised learning technique to identify groups with similar characteristics in the same group [23]. Agglomerative hierarchical clustering was used in this research. The bottom-up hierarchical algorithm treats each sample as a single cluster and then combines pair of clusters that are most similar until every cluster is grouped into one single cluster. In the case of general cross-section data, block distance, Euclidean distance, Minkowski distance, Chebychev distance, or Mahalanobis distance are used to measure the distance between two vectors $\left( \underline{x}_i' = [x_{i1}, x_{i2}, \ldots, x_{ip}] \right)$ and $\underline{x}_j' = [x_{j1}, x_{j2}, \ldots, x_{jp}] \right)$.

Cluster analysis of samples collected from multivariate panel data is often averaged over time data into general cross-section data. Typical Euclidean distance is then calculated for further grouping. However, this will result in information loss because the mean shows the average change in the data but does not show the distributing characteristics of the data, such as the standard deviation. Therefore, in this study, a Euclidean distance timed

and spaced ($d_{rk}$) is used to calculate the distance between sample *r* and sample *k* [15–18], as in Equation (3).

$$d_{rk} = \sqrt{\sum_{t=1}^{T} \sum_{j=1}^{p} \left( x_{rj}(t) - x_{kj}(t) \right)^2}, \tag{3}$$

The distance should satisfy some conditions as follows:

1. $d_{rk} \geq 0$, *if* $x_{rj}(t) = x_{kj}(t)$ *then* $d_{rk} = 0$
2. $d_{rk} = d_{kr}$, *to all* $x_{rj}(t), x_{rj}(t)$
3. $d_{rk} \leq d_{rl} + d_{kl}$, *to all* $x_{rj}(t), x_{kj}(t), x_{lj}(t)$

A distance matrix for spatial grouping analysis contains a distance value between every pair of samples as in Equation (4a), which is the symmetric matrix ($n \times n$) with all diagonal values of zero. At the same time, a distance matrix for temporal grouping analysis within the spatial cluster contains a distance value between every pair of months as in Equation (4b), which is the asymmetric matrix ($12 \times 12$) with all diagonal values of zero.

$$\begin{bmatrix} 0 & & & & \\ d_{21} & 0 & & & \\ d_{31} & d_{32} & 0 & & \\ \vdots & \vdots & \vdots & \ddots & \\ d_{n1} & d_{n2} & \cdots & d_{n(n-1)} & 0 \end{bmatrix}, \tag{4a}$$

$$\begin{bmatrix} 0 & & & & \\ d_{21} & 0 & & & \\ d_{31} & d_{32} & 0 & & \\ \vdots & \vdots & \vdots & \ddots & \\ d_{121} & d_{T2} & \cdots & d_{12(11)} & 0 \end{bmatrix}, \tag{4b}$$

Average linkage, which is the unweighted pair group method using arithmetic averages (UPGMA), was used to average the distance values between pairs of clusters [24]. It is widely used because it compromises the extreme cases [25].

The multivariate cluster analysis used in this paper was implemented directly using the "philanthropy", "cluster", "factoextra" and "FactoMineR" package in R programming language and RStudio [26].

### 2.4.2. Cluster Validation

This paper employed silhouette width ($S_i$) [27] to determine the optimal number of clusters, and it also could be used to validate consistency within clusters of data. The silhouette measures the similarity of *i*-th observation to its own cluster and the similarity of observation to other clusters as Equation (5).

$$S_i = \frac{b_i - a_i}{\max(b_i, a_i)} \tag{5}$$

where $a_i$ is the average distance between *i* and all other observations in the same cluster, and $b_i$ is the average distance between *i* and the observations in the "nearest neighboring cluster" as Equation (6).

$$b_i = \min_{C_k \in \mathcal{C}, \backslash C(i)} \sum_{j \in C_k} \frac{d(i,j)}{n(C_k)} \tag{6}$$

where $C(i)$ is the cluster containing observation *i*, $d(i,j)$ is the Euclidean distance timed and spaced between observations *i* and *j*, and $n(C)$ is the cardinality of cluster *C*.

$S_i$ ranges from $-1$ to $+1$, where a high value indicates that the observation is well matched to its own cluster, while a low or negative value indicates that observation is poorly matched to its own cluster. The average of observation's silhouette in a cluster was obtained to determine whether the clustering configuration is appropriate. The advantage

of using silhouette only depends on the actual partition of the observations, not on the clustering algorithm that was used, and no need to access the original data. This paper implemented this function using the silhouette function in package cluster [28].

## 3. Results

This section may be divided into subheadings. It should provide a concise and precise description of the experimental results, their interpretation, as well as the experimental conclusions that can be drawn.

### 3.1. Variable Characteristics

Figure 4 shows boxplots of seven variables; they are varied by month but have the same pattern each year.

Rainfalls were more varied than others in 1997 and 2007 for the El Niño and La Niña phenomenon, respectively. The average rainfall in La Niña phenomenon was higher than that in El Niño phenomenon and the normal average, except for 1999, which was affected by the 1997–1998 very strong El Niño. Furthermore, all factors in each year had a pattern in relation to the season. For example, rainfall was very high and more fluctuated from August to September. It can be concluded that climate factors were different from month to month and year to year. Obviously, the rainfall between El Niño and La Niña differed significantly, while other climate factors were similar. This suggested the rainfall should be more focused to analyse the impact of the ENSO phenomenon on spatial clustering.

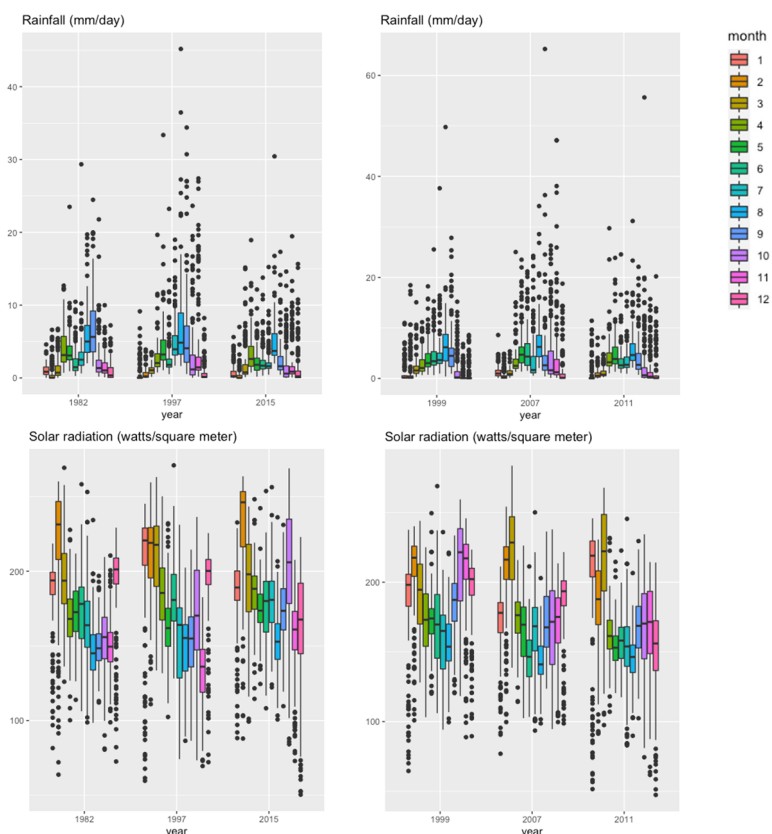

**Figure 4.** *Cont*.

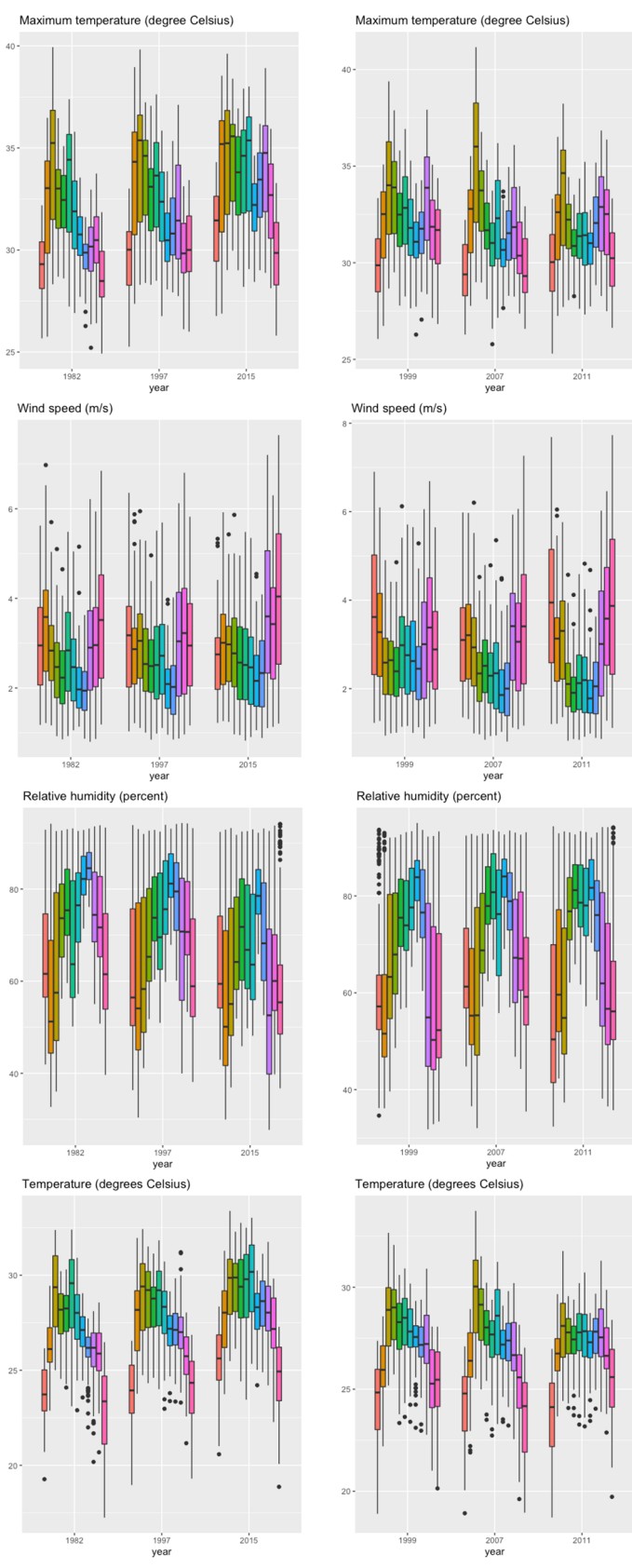

**Figure 4.** *Cont.*

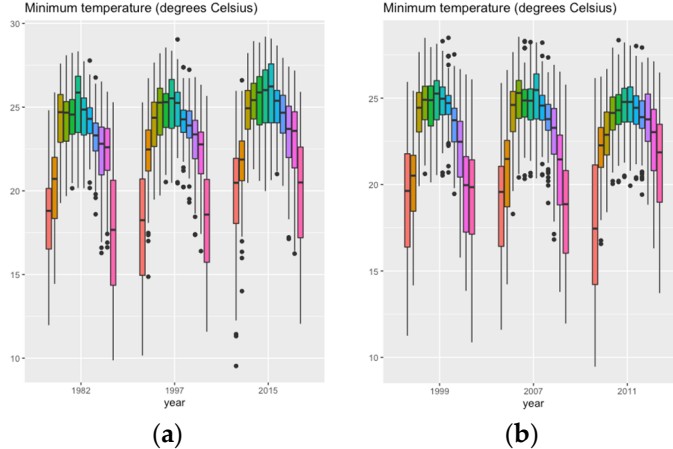

**Figure 4.** Boxplots of climate factors of 124 Thai meteorological stations by ENSO, years and months: (**a**) El Niño; (**b**) La Niña.

### 3.2. Spatial Clustering

The average silhouette width was used to determine a suitable number of clusters (*k*). It suggested the value 4 or 5 for *k*, due to their maximum width (Figure 5). So, a fair comparison between the ENSO events was achieved for choosing five spatial clusters (SCs) close to height 12.5 (distance between clusters) for all datasets in this study.

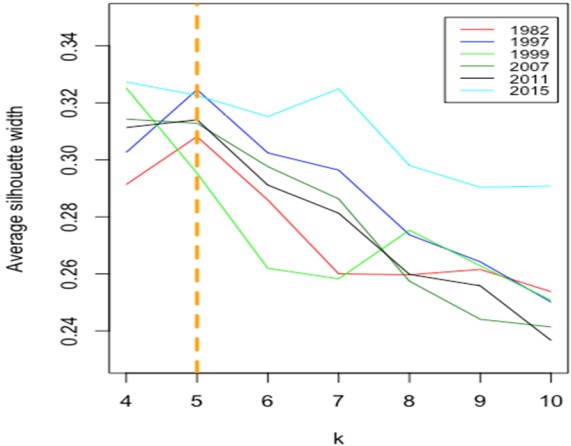

**Figure 5.** The average silhouette width for spatial cluster analysis by number of clusters and years.

Five spatial clusters, SC1, SC2, SC3, SC4, and SC5, which were sorted according to the amount of precipitation from ascending to high, were formed and displayed on a spatial map in Figure 6. It was obvious that precipitation was the only meteorological data to noticeably differ between clusters. Spatial clustering in El Niño events was mostly grouped in SC2 (yellow) with 62–66 members except in 1982, which mostly in SC1 (red) with 59 members; however, its average rainfalls were nearly the same to SC2, whereas spatial clustering in La Niña events was mostly grouped in SC1 (red) with 61–83 members. While SC5 (pink) was the least populated member with one member, which was the station in the east for both events (Table 3). These showed most areas in Thailand had low precipitation rate.

**El Niño**

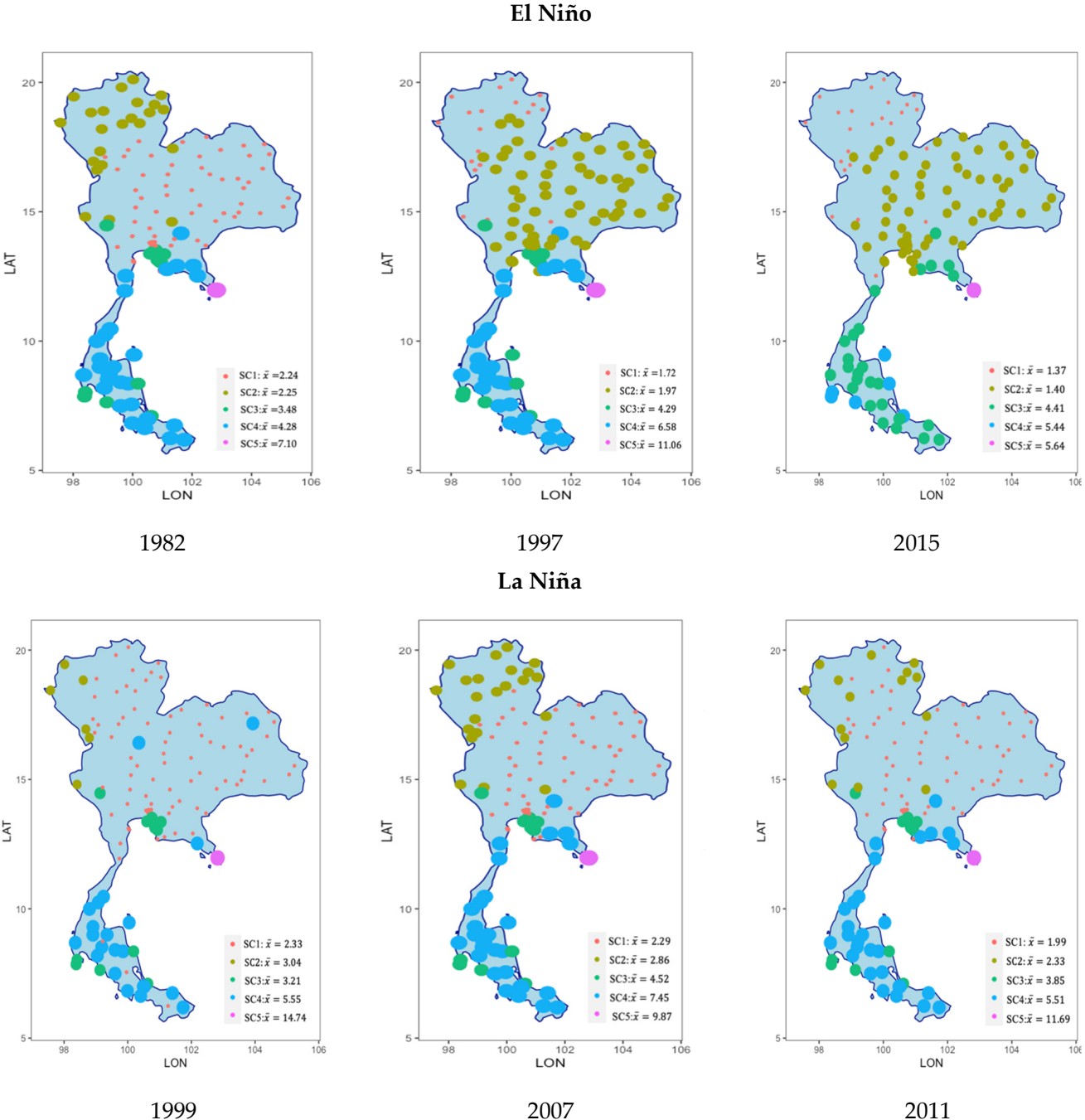

**Figure 6.** Spatial cluster analysis (SC1–SC5) for the 124 Thai meteorological stations on a map by ENSO events and years ($\bar{x}$ is monthly rainfall average).

**Table 3.** Number of members and mean of monthly climate factors by ENSO events, years and SCs.

| Year | SC | $n$ | C (n = 26) | E (n = 15) | N (n = 16) | NE (n = 28) | S (n = 27) | W (n = 12) | Rainfall | Relative Humidity | Solar Down Welling | Temperature | Max Temperature | Min Temperature | Horizontal Wind |
|---|---|---|---|---|---|---|---|---|---|---|---|---|---|---|---|
| | | | | | | | **El Niño** | | | | | | | | |
| 1982 | 1 | 59 | 23 | 5 | 1 | 26 | - | 4 | 2.24 | 64.56 | 178.31 | 27.47 | 32.46 | 23.01 | 3.08 |
| | 2 | 23 | 1 | - | 15 | 2 | - | 5 | 2.25 | 69.52 | 182.24 | 25.11 | 31.42 | 19.82 | 1.97 |
| | 3 | 12 | 2 | 4 | - | - | 5 | 1 | 3.48 | 75.41 | 188.43 | 27.26 | 29.32 | 25.38 | 4.14 |
| | 4 | 29 | - | 5 | - | - | 22 | 2 | 4.28 | 87.29 | 142.18 | 25.78 | 29.74 | 22.79 | 1.99 |
| | 5 | 1 | - | - | 1 | - | - | - | 7.10 | 78.55 | 163.36 | 25.90 | 28.25 | 24.00 | 3.18 |
| | Total $\bar{x}$ | | | | | | | | 2.88 | 71.96 | 171.45 | 26.60 | 31.30 | 22.60 | 2.72 |
| | $s$ | | | | | | | | 3.23 | 14.26 | 34.31 | 2.44 | 2.56 | 3.24 | 1.11 |
| 1997 | 1 | 20 | 1 | - | 12 | 2 | - | 5 | 1.72 | 66.95 | 191.63 | 25.64 | 32.24 | 20.07 | 2.07 |
| | 2 | 62 | 23 | 5 | 4 | 26 | - | 4 | 1.97 | 62.51 | 184.51 | 28.06 | 33.29 | 23.30 | 3.07 |
| | 3 | 13 | 2 | 4 | - | - | 6 | 1 | 4.29 | 76.00 | 183.52 | 27.54 | 29.60 | 25.67 | 4.00 |
| | 4 | 28 | - | 5 | - | - | 21 | 2 | 6.58 | 87.91 | 136.45 | 25.96 | 29.87 | 23.00 | 1.85 |
| | 5 | 1 | - | - | 1 | - | - | - | 11.06 | 79.55 | 149.37 | 26.18 | 28.38 | 24.37 | 3.32 |
| | Total $\bar{x}$ | | | | | | | | 3.29 | 70.51 | 174.42 | 27.13 | 31.92 | 22.97 | 2.73 |
| | $s$ | | | | | | | | 4.60 | 14.76 | 39.19 | 2.31 | 2.65 | 3.15 | 1.10 |
| 2015 | 1 | 24 | 1 | - | 15 | 2 | - | 6 | 1.37 | 60.61 | 195.98 | 26.79 | 33.67 | 20.81 | 2.12 |
| | 2 | 66 | 25 | 9 | 1 | 26 | - | 5 | 1.40 | 57.90 | 191.31 | 29.24 | 34.31 | 24.50 | 3.48 |
| | 3 | 27 | - | 5 | - | - | 21 | 1 | 4.41 | 86.58 | 140.25 | 26.66 | 30.76 | 23.55 | 1.92 |
| | 4 | 6 | - | - | - | - | 6 | - | 5.44 | 80.10 | 163.10 | 27.84 | 29.45 | 26.47 | 3.60 |
| | 5 | 1 | - | 1 | - | - | - | - | 5.64 | 76.31 | 167.87 | 26.99 | 29.60 | 24.91 | 3.21 |
| | Total $\bar{x}$ | | | | | | | | 2.28 | 65.89 | 179.54 | 28.12 | 33.14 | 23.68 | 2.88 |
| | $s$ | | | | | | | | 2.85 | 15.93 | 36.99 | 2.34 | 2.73 | 3.02 | 1.22 |
| | | | | | | | | **La Niña** | | | | | | | | |
| 1999 | 1 | 83 | 23 | 9 | 13 | 27 | 3 | 8 | 2.33 | 62.44 | 192.34 | 27.38 | 33.02 | 22.31 | 2.99 |
| | 2 | 6 | - | - | 3 | - | - | 3 | 3.04 | 73.09 | 184.80 | 24.29 | 31.00 | 19.03 | 1.83 |
| | 3 | 12 | 2 | 4 | - | - | 5 | 1 | 3.21 | 72.95 | 197.32 | 27.81 | 30.03 | 25.71 | 4.41 |
| | 4 | 22 | 1 | 1 | - | 1 | 19 | - | 5.55 | 88.22 | 142.53 | 26.12 | 30.09 | 23.13 | 1.99 |
| | 5 | 1 | - | 1 | 1 | - | - | - | 14.74 | 76.97 | 161.99 | 26.40 | 28.80 | 24.42 | 3.79 |
| | Total $\bar{x}$ | | | | | | | | 3.12 | 68.66 | 183.38 | 27.04 | 32.08 | 22.64 | 2.90 |
| | $s$ | | | | | | | | 3.89 | 16.07 | 34.45 | 2.09 | 2.28 | 3.16 | 1.16 |
| 2007 | 1 | 61 | 23 | 6 | 2 | 26 | - | 4 | 2.29 | 64.40 | 181.10 | 27.71 | 32.82 | 23.08 | 3.05 |
| | 2 | 22 | 1 | - | 14 | 2 | - | 5 | 2.86 | 70.56 | 187.98 | 25.10 | 31.40 | 19.78 | 1.96 |
| | 3 | 12 | 2 | 4 | - | - | 5 | 1 | 4.52 | 74.00 | 184.46 | 27.87 | 29.99 | 25.89 | 4.16 |
| | 4 | 28 | - | 4 | - | - | 22 | 2 | 7.45 | 87.43 | 139.07 | 26.22 | 30.18 | 23.21 | 1.91 |
| | 5 | 1 | - | - | 1 | - | - | - | 9.87 | 77.86 | 162.88 | 26.42 | 28.75 | 24.43 | 3.33 |
| | Total $\bar{x}$ | | | | | | | | 3.83 | 71.73 | 173.01 | 26.92 | 31.67 | 22.81 | 2.71 |
| | $s$ | | | | | | | | 5.38 | 14.20 | 35.00 | 2.33 | 2.46 | 3.15 | 1.12 |
| 2011 | 1 | 67 | 23 | 5 | 7 | 26 | - | 6 | 1.99 | 62.83 | 177.59 | 27.60 | 32.78 | 22.93 | 3.00 |
| | 2 | 15 | 1 | - | 9 | 2 | - | 3 | 2.33 | 68.55 | 181.98 | 24.91 | 31.09 | 19.76 | 1.95 |
| | 3 | 12 | 2 | 4 | - | - | 5 | 1 | 3.85 | 74.46 | 178.97 | 27.75 | 29.76 | 25.84 | 4.22 |
| | 4 | 29 | - | 5 | - | - | 22 | 2 | 5.51 | 87.37 | 132.16 | 26.20 | 29.91 | 23.45 | 2.03 |
| | 5 | 1 | - | 1 | - | - | - | - | 11.69 | 77.86 | 155.50 | 26.31 | 28.65 | 24.33 | 3.21 |
| | Total $\bar{x}$ | | | | | | | | 3.11 | 70.51 | 167.45 | 26.95 | 31.58 | 22.96 | 2.76 |
| | $s$ | | | | | | | | 3.96 | 15.88 | 37.61 | 1.88 | 2.16 | 2.92 | 1.34 |

$n$—number of members, C—Central, E—East, N—North, NE—Northeast, S—South, W—West.

In La Niña event, SC1 (red) was found mostly in Northeast and Central areas, which had the least amount of rainfall, and SC2 (yellow) was widely distributed in the North, which had low rainfall. SC3 (green) with moderate rainfall were distributed among all regions, except North and Northeast, while SC4 (blue) are in the south which had quite a lot of rainfall. Lastly, SC5 (pink) with the highest rainfall had one station in the East (Table 3).

While, spatial clustering in El Niño was differently distributed by years. In 1997 and 2015, SC1 (red) was found mostly in the North and SC2 (yellow) was widely distributed in the Northeast, and vice versa in 1982. In 1982 and 1997, SC3 (green) with moderate rainfalls were distributed among all regions, except North and Northeast, and SC4 (blue) were in the South which had quite a lot of rainfall, and vice versa in 2015. In every year, SC5 (pink) with the highest rainfall had one station in the East (Table 3).

The spatial clistering extracted the drought areas in the North region, classified as SC1 with less rainfall than SC2 and in the South region classified as SC3 with less rainfall than SC4 for the El Niño event. These areas would be at risk to be the most drought-prone areas. This suggested the effect of ENSO on spatial clustering.

The distribution of SGs over six regions, showing a clear trend in the redistribution of SGs observed in this study, is shown in Table 4. More diverse climate was found in the East and West than other regions. All regions had a heterogenous meteorological distribution. Every year for both El Niño and La Niña events had 2–4 SCs. However, less distribution for El Niño (2015) in Central, East, and West regions and for La Niña (1999) in the West, and more distribution for La Niña (1999) in the South were noted. These would be due to changes in TGs and intensity of climate factors.

**Table 4.** The distribution of SGs over six regions for ENSO.

| Region | Number of Members | El Niño | | | La Niña | | |
|---|---|---|---|---|---|---|---|
| | | **1982** | **1997** | **2015** | **1999** | **2007** | **2011** |
| Central | 26 | 3 | 3 | 2 | 3 | 3 | 3 |
| East | 15 | 4 | 4 | 3 | 4 | 4 | 4 |
| North | 16 | 2 | 2 | 2 | 2 | 2 | 2 |
| Northeast | 28 | 2 | 2 | 2 | 2 | 2 | 2 |
| South | 27 | 2 | 2 | 2 | 3 | 2 | 2 |
| West | 12 | 4 | 4 | 3 | 3 | 4 | 4 |

### 3.3. Temporal Clustering

After spatial cluster analysis had been obtained, a Euclidean distance timed and spaced with average linkage was next applied to the monthly climate factors for each SC to find temporal clusters (TCs) within each SG. Normally, Thailand has three seasons, summer (February–May), rainy (May–October), and winter (October–February). To compare temporal clusters of the ENSO phenomenon, three TCs within each SC were compared in this study. TC1, TC2, and TC3, which were sorted according to the amount of ascending precipitation, were represented by orange, blue, and green, respectively. TCs corresponding to each SG is shown in the dendrogram to depict the groups of clusters and their combination, indicating dissimilarity in the vertical scale and the samples (months) in clustering order on the horizontal axis. They help to see how long each season lasts and the different period of seasons in each spatial grouping (Figure 7).

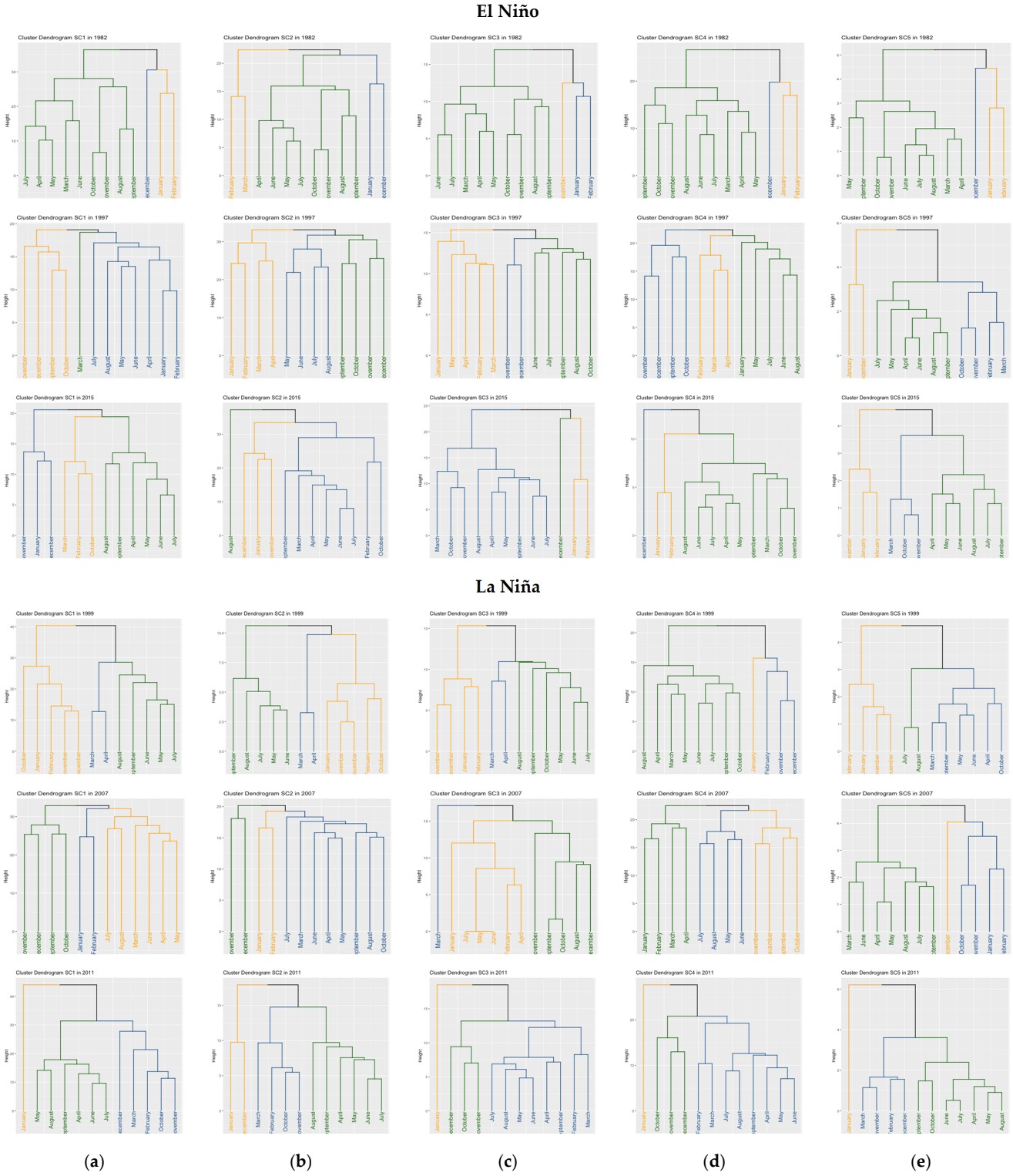

**Figure 7.** Dendrograms of TGs for the five different SCs discovered by ENSO events, years and SCs: (**a**) SC1; (**b**) SC2; (**c**) SC3; (**d**) SC4; (**e**) SC5.

For example, in 1982, TC1 and TC2 in SC1 depicted a very dry season with average precipitation intensity of less than 2 mm/day (Table 5). They were composed of three months. Months of TC1 were December and of TC2 were January and February. TC3, on the other hand, was a slightly wet season with an average precipitation of 2 mm/day or more for 9 months, March–November.

**Table 5.** Number of members (months) and mean of monthly climate factors by ENSO events, years, SCs and TCs.

| Year | SCs | | TC1 | | | | | | TC2 | | | | | | TC3 | | | | | |
| | SC | *n* | *n* | RF | RH | SDW | Temp | Wind | *n* | RF | RH | SDW | Temp | Wind | *n* | RF | RH | SDW | Temp | Wind |
|---|---|---|---|---|---|---|---|---|---|---|---|---|---|---|---|---|---|---|---|---|
| | | | | | | | | | **El Niño** | | | | | | | | | | | |
| 1982 | 1 | 59 | 2 | 0.45 | 51.90 | 217.32 | 25.64 | 3.53 | 1 | 0.64 | 56.18 | 203.21 | 22.77 | 3.91 | 9 | 2.82 | 68.30 | 166.87 | 28.39 | 2.89 |
| | 2 | 23 | 2 | 0.33 | 47.45 | 233.93 | 27.23 | 2.18 | 2 | 0.41 | 60.00 | 199.95 | 20.75 | 2.03 | 8 | 3.19 | 77.42 | 164.89 | 25.67 | 1.90 |
| | 3 | 12 | 1 | 0.78 | 67.94 | 199.38 | 25.38 | 5.82 | 2 | 1.08 | 70.43 | 202.37 | 26.12 | 5.22 | 9 | 4.31 | 77.35 | 184.12 | 27.72 | 3.71 |
| | 4 | 29 | 2 | 1.91 | 85.07 | 145.13 | 24.52 | 2.63 | 1 | 2.24 | 84.68 | 139.28 | 24.04 | 2.92 | 9 | 5.03 | 88.07 | 141.84 | 26.26 | 1.75 |
| | 5 | 1 | 2 | 1.21 | 68.36 | 216.51 | 24.63 | 4.14 | 1 | 4.50 | 71.78 | 206.18 | 23.65 | 5.33 | 9 | 8.70 | 81.57 | 146.79 | 26.43 | 2.72 |
| 1997 | 1 | 20 | 4 | 1.53 | 66.75 | 193.05 | 25.77 | 2.02 | 7 | 1.78 | 67.14 | 190.66 | 25.57 | 2.09 | 1 | 2.07 | 66.46 | 192.71 | 25.56 | 2.24 |
| | 2 | 62 | 4 | 1.91 | 62.18 | 185.43 | 28.02 | 3.03 | 4 | 1.97 | 62.49 | 184.67 | 28.11 | 3.11 | 4 | 2.02 | 62.85 | 183.43 | 28.04 | 3.06 |
| | 3 | 13 | 5 | 3.38 | 74.61 | 188.82 | 27.43 | 3.96 | 2 | 3.82 | 76.89 | 186.10 | 27.56 | 4.19 | 5 | 5.39 | 77.02 | 177.19 | 27.64 | 3.95 |
| | 4 | 28 | 3 | 6.42 | 87.73 | 136.62 | 25.90 | 1.83 | 4 | 6.49 | 87.92 | 134.76 | 25.97 | 1.84 | 5 | 6.75 | 88.00 | 137.71 | 25.99 | 1.87 |
| | 5 | 1 | 2 | 0.01 | 70.45 | 221.12 | 24.34 | 5.25 | 4 | 4.12 | 77.73 | 163.98 | 26.08 | 3.25 | 6 | 19.37 | 83.80 | 115.71 | 26.87 | 2.72 |
| 2015 | 1 | 24 | 3 | 0.26 | 45.56 | 237.11 | 27.97 | 2.23 | 3 | 0.34 | 54.87 | 192.98 | 23.42 | 2.21 | 6 | 2.44 | 71.01 | 176.91 | 27.88 | 2.01 |
| | 2 | 66 | 3 | 0.44 | 54.75 | 176.82 | 26.57 | 3.98 | 8 | 1.33 | 56.87 | 200.20 | 30.27 | 3.42 | 1 | 4.88 | 75.60 | 163.62 | 28.95 | 2.52 |
| | 3 | 27 | 2 | 1.79 | 85.62 | 151.94 | 25.27 | 2.08 | 9 | 4.88 | 87.07 | 142.13 | 27.14 | 1.74 | 1 | 5.49 | 84.11 | 100.02 | 25.09 | 3.15 |
| | 4 | 6 | 2 | 2.13 | 78.12 | 179.29 | 26.77 | 4.22 | 1 | 5.84 | 82.57 | 108.83 | 26.40 | 5.88 | 9 | 6.14 | 80.27 | 165.54 | 28.24 | 3.21 |
| | 5 | 1 | 3 | 0.30 | 69.08 | 203.24 | 25.66 | 4.22 | 3 | 1.33 | 71.78 | 185.00 | 27.42 | 3.68 | 6 | 10.46 | 82.20 | 141.63 | 27.44 | 2.46 |
| | | | | | | | | | **La Niña** | | | | | | | | | | | |
| 1999 | 1 | 83 | 5 | 0.17 | 49.95 | 213.89 | 25.78 | 3.36 | 2 | 1.86 | 61.83 | 187.93 | 29.51 | 2.70 | 5 | 4.68 | 75.17 | 172.57 | 28.13 | 2.74 |
| | 2 | 6 | 5 | 0.11 | 59.90 | 219.48 | 22.68 | 1.87 | 2 | 1.16 | 65.64 | 196.45 | 27.46 | 1.62 | 5 | 6.73 | 89.27 | 145.45 | 24.64 | 1.87 |
| | 3 | 12 | 4 | 1.69 | 65.81 | 199.37 | 26.91 | 5.33 | 2 | 3.70 | 77.43 | 184.96 | 28.20 | 3.54 | 6 | 4.06 | 76.22 | 200.08 | 28.27 | 4.08 |
| | 4 | 22 | 1 | 4.61 | 89.13 | 103.48 | 24.90 | 2.97 | 3 | 4.77 | 87.38 | 140.23 | 25.08 | 2.39 | 8 | 5.96 | 88.42 | 148.28 | 26.67 | 1.71 |
| | 5 | 1 | 4 | 0.85 | 64.87 | 225.47 | 25.49 | 4.78 | 6 | 14.34 | 81.83 | 141.38 | 26.80 | 2.92 | 2 | 43.70 | 86.56 | 96.88 | 27.01 | 4.43 |
| 2007 | 1 | 61 | 6 | 2.22 | 64.76 | 181.64 | 27.62 | 3.10 | 2 | 2.33 | 63.08 | 181.16 | 27.88 | 3.04 | 4 | 2.38 | 64.51 | 180.27 | 27.76 | 2.98 |
| | 2 | 22 | 2 | 2.41 | 70.04 | 188.90 | 25.29 | 1.95 | 8 | 2.95 | 70.70 | 186.97 | 25.02 | 1.95 | 2 | 2.95 | 70.53 | 191.07 | 25.26 | 2.01 |
| | 3 | 12 | 6 | 2.12 | 70.71 | 197.18 | 28.12 | 4.51 | 1 | 2.83 | 69.34 | 211.15 | 26.99 | 4.00 | 5 | 7.74 | 78.87 | 163.87 | 27.76 | 3.76 |
| | 4 | 28 | 4 | 7.09 | 87.45 | 140.04 | 26.20 | 1.89 | 4 | 7.19 | 87.33 | 139.16 | 26.24 | 1.91 | 4 | 8.06 | 87.52 | 138.01 | 26.23 | 1.92 |
| | 5 | 1 | 1 | 0.06 | 64.49 | 215.43 | 24.41 | 5.29 | 4 | 2.51 | 72.80 | 196.31 | 25.93 | 4.26 | 7 | 15.47 | 82.66 | 136.28 | 26.99 | 2.52 |
| 2011 | 1 | 67 | 1 | 0.02 | 45.31 | 222.12 | 23.50 | 4.01 | 5 | 0.43 | 52.37 | 188.69 | 27.67 | 3.72 | 6 | 3.61 | 74.47 | 160.92 | 28.22 | 2.23 |
| | 2 | 15 | 2 | 0.03 | 50.35 | 210.01 | 21.59 | 2.51 | 4 | 0.27 | 54.41 | 210.47 | 25.88 | 2.16 | 6 | 4.47 | 84.04 | 153.64 | 25.37 | 1.63 |
| | 3 | 12 | 1 | 1.66 | 65.51 | 186.20 | 25.98 | 6.57 | 8 | 3.84 | 77.03 | 184.95 | 27.93 | 3.56 | 3 | 4.62 | 70.56 | 160.61 | 27.86 | 5.20 |
| | 4 | 29 | 1 | 3.77 | 84.22 | 117.80 | 24.59 | 3.30 | 8 | 5.00 | 87.95 | 142.08 | 26.45 | 1.69 | 3 | 7.42 | 86.86 | 110.50 | 26.10 | 2.50 |
| | 5 | 1 | 1 | 0.10 | 67.75 | 214.62 | 24.21 | 6.09 | 4 | 1.28 | 71.65 | 182.57 | 26.23 | 3.96 | 7 | 19.29 | 82.86 | 131.58 | 26.66 | 2.38 |

*n*—number of members, RF: rainfall, RH: relative humidity, SWD: solar downwelling, Temp: temperature, Wind: horizontal wind.

TC1 and TC2 in SC2, 1982 depicted a very dry season with average precipitation intensity of less than 2 mm/day. TC1 was composed of February–March. TC2 was December and January. TC3, on the other hand, was a wet season with an average precipitation of 4.51 mm/day for 8 months, April–November.

TC1 and TC2 in SC3, 1982 was a dry season for three months, December (TC1) and January–February (TC2), with an average precipitation of less than 2 mm/day, while TC3 was a wet season with an average precipitation of 4 mm/day or more for 9 months, March–November.

TCs in SC4 and SC5, 1982 were the same. TC1 was a dry season, January–February, with an average precipitation of less than 2 mm/day, while TC2 and TC3 were a wet season with an average precipitation of 2 mm/day or more for 10 months, December (TC2) and March–November (TC3).

TCs in each SC in La Niña were similar to those in El Niño. Nevertheless, there was higher average precipitation intensity in La Niña phenomenon, than those in El Niño phenomenon. Furthermore, the rainy season was a longer period in SC4 and SC5 for both events of ENSO.

## 4. Discussion

The highest average rainfall in 1982, 1997, and 2015 (5.64–11.06 mm/day) was less than that of in 1999, 2007, and 2011 (9.87–14.74 mm/day). This corresponds to the Oceanic Niño Index (ONI), showing that ONI in 1982, 1997, and 2015 was greater than 0.5 °C, meaning that El Niño occurred, and in 1999, 2007, and 2011 was below −0.5 °C, meaning that La Niña occurred [29].

Lower rainfall than usual was found, so there was a widespread drought in almost all regions of Thailand in 1982 and 1997, especially in Northeast [30]. There also was a severe El Niño effect in 2015, causing very low precipitation across the country ($\bar{x} = 2.28$ mm/day).

Five spatial clusterings were formed. SC5 with the highest average precipitation was formed by only one station in Khlong Yai District, Trat Province, in every year whether there was an El Niño or La Niña phenomenon ($\bar{x} = 5.64 - 14.74$ mm/day). The topography of Khlong Yai District is a coastline fully influenced by the southwest monsoon from the Gulf of Thailand; consequently, it has abundant rainfall for most of the year. This is consistent with the Trat Agricultural Meteorological Document that reports that Khlong Yai District, Trat Province, is the wettest area in Thailand [31].

There were approximately 80 stations in SC1 and SC2 with low average precipitation and especially low in 2015, mostly in the Central, North, and Northeast. It was consistent with a report that rainfall in these three regions when El Niño occurred was less than the average 30 years of rainfall of normal years.

There were three TCs in each SC. When the El Niño phenomenon occured, Thailand rainfall tended to be lower than normal, especially during the summer and early rainy season (mid-February–June). The dry season in El Niño was longer and less than average rainfall than TCs for the La Niña phenomena.

Most stations in the south were clustered into SC3 and SC4 with moderate and high rainfall, respectively, for both El Niño and La Niña phenomena. Usually, rainfall in Thailand, especially in the southeast coast, is high during October–December. In addition, some parts of Thailand were not affected by the ENSO phenomenon (El Niño and La Niña), such as Trat in SC5 with the highest rainfall, and Tak, Chiang Rai, Chiang Mai, Phayao, and Lampang in SC1 with the least rainfall. This may be due to their topography.

There are 35 provinces with more than one meteorological station of TMD. Of these, stations in 34 provinces were grouped into different SCs. This may be due to their topography affecting a different climate.

Spatial clusters were similar for both El Niño and La Niña except in 2015, when severe El Niño occurred. This might be the Euclidian distance matrix tending to cluster the samples with climate variables having similar mean. This suggests that other similarity matrices,

such as correlation, may be possible to group samples based on trends and variation over time [11].

## 5. Conclusions

This paper employed multivariate cluster analysis with the average linkage to analyze the spatial and temporal grouping, using climate factors which are rainfall, relative humidity, average temperature, maximum temperature, lowest temperature, solar radiation, and wind speed at 124 locations over Thailand from CCAM (10 km), for the years 1982, 1997, and 2015 (El Niño) and 1999, 2007, and 2011 (La Niña).

Five SCs with a distance between a cluster of 12.5 were compared. It was observed that SCs were similar for both El Niño and La Niña except in 2015, when severe El Niño occurred. This indicated the more severe El Niño, the more spatial variation. The main difference between SC1–SC5 was the ascending amount of precipitation, where SC1 had the least amount of rainfall and SC5 had the heaviest rainfall.

In addition, three TC patterns in each SC were similar for both El Niño and La Niña. Nevertheless, the average precipitation intensity in La Niña was higher than that in El Niño.

This paper implements cluster analysis on atmospheric panel data. Even multivariable panel data is more complicated, but it is practical to cluster. Cluster results arealso more realistic than cross-sectional data and avoid information loss.

Future studies may focus on using future climate factors from the weather forecast models for clustering to study the spatial and temporal distributions. Other than the correlation distance suggested, the robust distance, for example the absolute distance or the Canberra distance to deal with outliers, should be further studied. Furthermore, as there might be extreme whether events in the ENSO phenomenon, for example less or abundant precipitation, which may affect the clustering, outliers should be detected and handled prior.

**Author Contributions:** Conceptualization, P.D.; methodology, P.D.; software, P.D., N.J., P.Y. and S.P.; validation, P.D.; formal analysis, P.D., N.J., P.Y. and S.P.; investigation, P.D.; resources, P.D., N.J., P.Y. and S.P.; data curation, P.D., N.J., P.Y. and S.P.; writing—original draft preparation, P.D., N.J., P.Y. and S.P.; writing—review and editing, P.D., U.H. and S.I.; visualization, P.D.; supervision, P.D.; project administration, P.D., U.H. and S.I. All authors have read and agreed to the published version of the manuscript.

**Funding:** This research project was supported by the Agricultural Research De-velopment Agency (ARDA) [PRP6405031190] and Thailand Sciece Research and Innovation (TSRI). Basic Research Fund: Fiscal year 2022 under project number FRB650048/0164.

**Conflicts of Interest:** The authors declare no conflict of interest.

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
