# Peer review of "Multivariable Panel Data Cluster Analysis of Meteorological Stations in Thailand for ENSO Phenomenon"

_mca, doi:10.3390/mca27030037_

Round 1

Reviewer 1 Report

Reviewer’s Report on Manuscript mca-1593929

“Multivariable Panel Data Cluster

 Analysis of Meteorological Stations

in Thailand for ENSO Phenomenon”,

Submitted to Mathematical and Computational Applications

by Porntip Dechpichai, Nuttawadee Jinapang, Pariyakorn

Yamphli, Sakulrat Polamnuay, Sittisak Injan, Usa Humphries

Summary. The paper investigates the spatial and temporal clustering of 124 meteorological stations in Thailand to have a better understanding of the ENSO impact on climate changes. Cluster analysis is based on average link clustering with Euclidean distance and silhouette width is used to validate the clustering results. Five spatial clusters and three time clusters were found and their behavior under cyclones were analyzed. Their behaviors were quite similar but in one case.

Overview. The effects of extreme whether events are worth studying, especially for countries whose economy heavily depends on agriculture. The present paper addresses the problem using a three-way data matrix, where units (stations), variables (climate) and occasions (months) are conveniently arranged. The chosen clustering approach is a default one, with some limitations mostly related to the possible presence of outliers. The material is carefully presented but may benefit from minor changes. More detailed comments follow.

Outliers. The available data are strongly influenced by extreme whether events, which might be regarded as outliers, from a statistical point of view. The Euclidean distance is notoriously sensitive to outliers. Hence the statistical analysis needs account for outliers. The first step is using a more robust distance, as for example the absolute distance or the Canberra distance. The second step would be to detect outliers. In recent years, there has been a wealth of research devoted to projections purported to visualize outliers. A short overview of the related literature follows.

Literature. Since Peña and Prieto (2001), projections maximizing kurtosis showed their potential for detecting outliers. They might be used alone or together with random projections (Peña and Prieto, 2007). They have also been used to detect outliers in multivariate time series (Galeano et al, 2006; Loperfido, 2020). Alternatively, outliers might be detected by means of skewness-maximizing projections (Franceschini and Loperfido, 2019; Loperfido, 2018). The use of such methods falls outside the scope of the present paper but should at least mentioned as a future research direction.

Minor. There are some minor points that should be addressed to improve the quality of the presentations.

  1. Please explain what “ENSO” means.
  2. Page 5, line 153. “asymmetric” should be “symmetric”.
  3. Page 5 line 166. “observation (i)” should be “i-th observation”.
  4. Page 5, lines 175 and 177. “Silhouette” should be “silhouette”.

Additional references

Franceschini, C. and Loperfido, N. (2019). MaxSkew and MultiSkew, Two R Packages for Detecting, Measuring and Removing Multivariate Skewness. Symmetry 11 (8), 970.

Galeano, P., D. Peña, and R. S. Tsay. 2006. “Outlier Detection in Multivariate Time Series by Projection Pursuit.” Journal of the American Statistical Association 101: 654–669.

Loperfido, N. (2018). Skewness-Based Projection Pursuit: a Computational Approach. Computational Statistics and Data Analysis 120, 42-57.

Loperfido N. (2020). Kurtosis-Based Projection Pursuit for Outlier Detection in Financial Time Series. The European Journal of Finance 26, 142-164.

Peña, D., and F. J. Prieto. 2001. “Multivariate Outlier Detection and Robust Covariance Estimation (with Discussion).” Technometrics 43: 286–310.

Peña, D., and F. J. Prieto. 2007. “Combining Random and Specific Directions for Outlier Detection and Robust Estimation of High dimensional Multivariate Data.” Journal of Computational and Graphical Statistics 16, 228–254.

Author Response

Dear Reviewers 1

It would be appropriated to have your advice to make the research more complete. We have made the corrections as the following table

Best regards,

Research team.

Reviewer 2 Report

Dear Editor, dear Authors,

I think this paper needs to be thoroughly reworked for publication. It suffers from serious shortcomings in both method and purpose.

  • The purpose of the paper is unclear. There is confusion between climate change (which we imagine to be of anthropogenic origin) and ENSO. ENSO is a natural phenomenon. Thailand is strongly impacted due to the Walker circulation which is modified during ENSO. Precipitations are less abundant during El Nino, more abundant during La Nina.
  • The method consisting in using dendrograms to represent clusters applying to spatialized or temporal data is not relevant because it is indifferent to spatial or temporal continuity. Meteorological data at one station are generally correlated to neighboring stations. On the other hand, these data are generally subject to seasonality.
  • As expected, the results are disappointing because of a lack of sensitivity: Lines 305-307 “It was observed that SCs were similar for both El Niño and La Niña except in 2015, of which severe El Niño occurred.”
  • The authors point out definitions unnecessarily (mean, standard deviation). Tables of figures are illegible.
  • Some sentences are incomprehensible: Reformulate lines 68-70

For these reasons, the paper should be declined for publication.

Author Response

Dear Reviewers 2

It would be appropriated to have your advice to make the research more complete. We have made the corrections as the following table

Best regards,

Research team.

Reviewer 3 Report

The novelty is a key criterion for the selection of manuscripts to be considered for publication in the MCA journal. I have a couple of concerns with the presented methodology that should be addressed. The manuscript should be returned to the authors for a round of major revisions and re-evaluated afterward. However, more details about the introduction, data, model development, and optimization need to be clarified. My major comments and questions are as follows:

  • The introduction section is not clear. The authors discussed mostly weather information in the introduction section. Why do not use data merging ML/DL algorithm to Cluster Analysis of Meteorological Stations and compare your results? What is the reason for choosing your algorithms over other established ML/DL-based data assimilation techniques? The popular ML-based data assimilation techniques: Quantile regression forest, random forest, SVM, boosted tree, Neural Network, etc which are successfully applied in the meteorological forecast. What is the uniqueness of the proposed technique and its potential impacts, over other established techniques? There are tons of weather forecast-based data assimilation models such as Quantile regression forest, SVM, boosted tree, etc., which you need to introduce. I think the authors need to propose a detailed comprehensive introduction section. The authors should explain with a new paragraph on this aspect in the introduction section. Also, you need to provide more literature reviews in the introduction section associated with the research gap/limitation.
  • Can you provide a high impactful schematic diagram to understand the proposed algorithms where the big impact of the results can be presented?
  • Do you think these amount datasets are enough to construct a Cluster analysis? Please justify?
  • Again, it is important to see how significant the dataset is for this proposed method. Can you include a significance test in the analysis?
  • You used multiple datasets with different sources (meteorological) with different temporal resolutions. How do you merge all those datasets into a common time series, please explain? It is necessary to report how the matching is carried out. I suggest adding a table that will include information about the resolution(temporal/spatial), sources, data ranges, etc.
  • How do you optimize your model? Can you provide a schematic diagram to understand the model setup?
  • Please add a discussion section and compare your results with previously established results?

Author Response

Dear Reviewers 3

It would be appropriated to have your advice to make the research more complete. We have made the corrections as the following table

Best regards,

Research team.

Reviewer 4 Report

This paper presents a multivariable panel data cluster analysis of meteorological stations in Thailand for ENSO phenomenon. The paper reads well but has room for the improvement. See the below:

  • Abstract is fine but english checking (for example, line 16) is required.
  • Very less number of literature are studied. More literature review are necessary.
  • Figure 1: Remove the background grid.
  • Table 1: No line in between, only first and last.
  • Figures 2, 4,5: Same comment as Figure 1.
  • Discussion requires improvement. Very small paragraph.

Author Response

Dear Reviewers 4

It would be appropriated to have your advice to make the research more complete. We have made the corrections as the following table

Best regards,

Research team.

Round 2

Author Response

Dear Reviewers,

It would be appropriated to have your advice to make the research more complete. We have made the corrections as the following table.

Best regards,

Research team.

Reviewer 3 Report

The authors significantly improved the quality of the paper by addressing most of the previous comments. This research work will be very effective for the Water resources community. I recommend the manuscript for publication after minor updates:

  • Can you provide a table for the primary meta-parameters for the multivariable Panel Data Clustering framework which will provide a clear idea about your proposed network?

  • You are focusing on your study over the complex terrain. Over complex terrain regions with different climatic regions, hydraulic/hydrological estimates can be associated with significant error due to variability and uncertainty introduced by orographic effects (Derin et al. 2016; Mei, et al. 2016, Khan et al. 2021, and so on). The authors should include this aspect in the introduction section.

Khan, et al 2021: Artificial Intelligence-Based Techniques for Rainfall Estimation Integrating Multisource Precipitation Datasets. Atmosphere 202112, 1239. 

Derin, et al . 2016 "Evaluation of multiple satellite-based precipitation products over complex topography." Journal of Hydrometeorology 15.4 (2014): 1498-1516.

Mei, et al. 2016: Evaluating satellite precipitation error propagation in runoff simulations of mountainous basins. J. Hydrometeor., 17, 1407–1423, https://doi.org/10.1175/JHM-D-15-0081.1.

Author Response

(The authors gave the same response as above.)
